# 3,5-T2-an Endogenous Thyroid Hormone Metabolite as Promising Lead Substance in Anti-Steatotic Drug Development?

**DOI:** 10.3390/metabo12070582

**Published:** 2022-06-23

**Authors:** Rajas Sane, Eva K. Wirth, Josef Köhrle

**Affiliations:** 1Institut für Experimentelle Endokrinologie, Charité–Universitätsmedizin Berlin, Freie Universität Berlin and Humboldt-Universität zu Berlin, D-10115 Berlin, Germany; rajas.sane@charite.de; 2Department of Endocrinology and Metabolism, Charité–Universitätsmedizin Berlin, Freie Universität Berlin and Humboldt-Universität zu Berlin, D-10115 Berlin, Germany; eva.wirth@charite.de; 3DZHK (German Centre for Cardiovascular Research), Partner Site Berlin, D-10115 Berlin, Germany

**Keywords:** 3,5-diiodothyronine (3,5-T2), liver, heart, pituitary, mitochondria, lipid metabolism, obesity, hypothyroidism, high fat diet, thyromimetic effects, canonical actions, HPTP axis

## Abstract

Thyroid hormones, their metabolites, and synthetic analogues are potential anti-steatotic drug candidates considering that subclinical and manifest hypothyroidism is associated with hepatic lipid accumulation, non-alcoholic fatty liver disease, and its pandemic sequelae. Thyromimetically active compounds stimulate hepatic lipogenesis, fatty acid beta-oxidation, cholesterol metabolism, and metabolic pathways of glucose homeostasis. Many of these effects are mediated by T3 receptor β1-dependent modulation of transcription. However, rapid non-canonical mitochondrial effects have also been reported, especially for the metabolite 3,5-diiodothyronine (3,5-T2), which does not elicit the full spectrum of “thyromimetic” actions inherent to T3. Most preclinical studies in rodent models of obesity and first human clinical trials are promising with respect to the antisteatotic hepatic effects, but potent agents exhibit unwanted thyromimetic effects on the heart and/or suppress feedback regulation of the hypothalamus-pituitary-thyroid-periphery axis and the fine-tuned thyroid hormone system. This narrative review focuses on 3,5-T2 effects on hepatic lipid and glucose metabolism and (non-)canonical mechanisms of action including its mitochondrial targets. Various high fat diet animal models with distinct thyroid hormone status indicate species- and dose-dependent efficiency of 3,5-T2 and its synthetic analogue TRC150094. No convincing evidence has been presented for their clinical use in the prevention or treatment of obesity and related metabolic conditions.

## 1. Introduction

Thyroid hormones (THs), secreted by the thyroid gland, affect various physiological processes, including thermal homeostasis, energy metabolism, and oxygen consumption during both the developmental and adult stages of animals and humans. The thyroid gland mainly secretes thyroxine (T4), which lacks biological activity under physiological conditions. T4 is metabolized to the active 3,5,3′-triiodo-L-thyronine (T3) or the inactive 3,3′,5′-triiodothyronine (reverse T3 or rT3) via the intracellular action of deiodinase isoenzymes [1]. While the deiodination of T3 results in the formation of metabolically inactive 3,3′-diiodothyronine (3,3′-T2), it remains an open question whether the deiodination process of T3 forms the active metabolite 3,5-diiodothyronine (3,5-T2) [2]. The TH metabolite 3,5-T2 has received significant attention over the last few years [1]. The following narrative review addresses the initial discovery and research on 3,5-T2 and describes its effects on lipid and glucose metabolism on the HPTP axis, liver, and heart of the rodents, along with the impact this metabolite has on the mitochondria.

### 1.1. Initial Decades of Promising 3,5-T2 Research

Magnus-Levy was among the first to study the relationships between clinical conditions with altered TH status (e.g., myxedema, goitre, M. Basedow, obesity) and metabolism and reported that feeding of thyroid extract tablets (“Thyrojodin”) resulted in weight reduction, water loss, higher lipid turnover, and an increased respiratory quotient RQ (CO_2_/O_2_) in several patients [3,4] at a time when mitochondria were already known but their function in metabolism was still elusive. Already in 1924 detailed information on the calorigenic action of T4 in humans was available: *“ … the total excess heat production resulting from the intravenous administration of a known amount of thyroxin shows that for each milligram of thyroxin injected approximately 1008 calories are produced … in the thyroidless individual …”* [5].

More than 300 publications on 3,5-Diiodo-L-thyronine (3,5-T2) are listed in PubMed since the first reports appeared on this compound. In 1927, Gaddum et al. reported the first experimental evidence of the bioactivity of 3,5-T2 in the acceleration of tadpole metamorphosis and oxygen consumption (Figure 1) albeit at a potency more than one order of magnitude lower than that observed for thyroxine (T4). Furthermore, the bioactivity ratio of 3,5-T2 for the acceleration of tadpole metamorphosis vs. stimulation of oxygen consumption differed from the ratio observed for these two endpoints using T4 [6,7]. In 1933, Anderson et al. described successful treatment of six severely myxedematous patients with oral liquid doses of 3,5-T2 between 50–75 mg/day (Figure 2). These patients previously had been on oral doses of poorly standardized thyroid preparations and were non-compliant on this regimen or experienced typical side effects such as palpitations, headaches, giddiness, hair loss, etc. [8]. The poor solubility of synthetic T4 at that time prevented its regular use as a drug due to the problems with intestinal absorption and frequent induction of hyperthyroidism at typical 1 mg doses daily administered by injection. Surprisingly, oral 3,5-T2 treatment did not induce those unpleasant side effects provoked by oral thyroid preparations or injected T4. In all six cases 3,5-T2 treatment could *“be used successfully for the relief of the symptoms of myxoedema”* [8]. BMR, previously strongly reduced, returned to normal but not elevated levels, and slightly increased the resting pulse rate by about 10 beats/min to around 65–78 beats/min which was severely decreased before. Patients lost about 5 kg of weight during treatment periods (2–3 weeks); sensory perceptions, mood, speech, alertness, and activity improved markedly. No toxic symptoms were observed. The authors assessed the responses to the chosen oral 50 mg/d dose as *“indeed very similar to those which would be obtained with the daily injection of 1 mg of thyroxine, and there seems no reason to doubt that the 3,5-diiodothyronine is acting as a true substitute for the thyroid hormone.”* [8]. Whether the weight loss was only due to the elimination of accumulated water [9,10], or also fat and protein loss during the short treatment periods remains open. However, it is quite remarkable that increases in resting pulse rate and body temperature were modest, BMR increases did not exceed regular levels and neither palpitations nor characteristic signs of hyperthyroidism were observed as typically reported by patients under thyroid or T4 dosage (or even more so after triiodothyronine (T3) treatment following its discovery) [11]. Apparently, 3,5-T2 did not elicit the full spectrum of “thyromimetic” actions inherent to thyroid extracts, the prohormone T4, or the main thyromimetically active hormone T3. Harrington and Barger had previously noticed a higher solubility of 3,5-T2 allowing better standardization of TH dosing compared to T4. 3,5-T2, easy to synthesize and to iodinate, was the penultimate precursor in the first total synthesis of the poorly soluble T4 which Harington and Barger accomplished together in 1927 [12]. Effects of T4, 3,5-T2, T3, and other thyroid hormone metabolites (THM) showed considerable latency compared to those of catecholamines and marked differences between various target organs and species with trends to decreased latency with decreasing size of the experimental animals. These observations lead to speculations on their distribution, cellular entry, and need of metabolic activation prior to their rapid or delayed, direct or indirect action on metabolism, growth, and energy expenditure.

Gemill et al. suggested that T4 and 3,5-T2 generate free radicals which interfere with ascorbic acid oxidation. This biochemical reaction was used as in vitro model to study the metabolic action of T4 [13]. Thermogenic and anti-goitrogenic effects of 3,5-T2 in rats were shown early on [14] in comparison with T4 where 3,5-T2 was the only T4 analogue with modest activities (7–11 % vs. T4), tested in that series that lacked substituents at the phenolic ring.

### 1.2. Discovery of 3,5-T2 in the Serum and Subsequent Research

In 1978, the biochemical characterization of three deiodinase (DIO) isoenzymes and the systematic development of highly specific radioimmunoassays (RIA) for various THs and their metabolites by several groups resulted in the identification of 3,5-T2 as endogenous TH metabolite in the blood of healthy individuals [2]. Alterations of 3,5-T2 concentrations were detected under clinically relevant (patho-)physiological conditions. Subsequent research culminated in the observation that 3,5-T2 rapidly stimulated hepatic oxygen consumption [15], probably mediated by a direct action on mitochondria independent of canonical TR-mediated signaling. This finding triggered an avalanche of reports on 3,5-T2 acting as a potential endogenous or pharmacologically useful TH metabolite for prevention or treatment of metabolic diseases such as obesity, diabetes, or the metabolic syndrome. Numerous studies followed, employing various animal experimental models from poikilothermic fish to warm-blooded rodents, experimental paradigms of thermoregulation or adaptation to altered environmental and nutritive conditions, especially under a hypothyroid state.

In 1989, Horst et al. demonstrated that the potent Dio1 inhibitor PTU prevented stimulatory T3 effects on respiration by using PTU during in vitro perfusion of livers removed after an overnight fast from radioiodine pre-treated hypothyroid rats [15]. However, earlier experiments revealed a remarkable stability of T3 over 2 h without evidence of a significant production of 3,3′-T2 using an in vitro recirculating perfusion model of euthyroid rat livers (ca. 10 g wet weight) in a similar experimental setup, but with a fully synthetic fluorocarbon medium acting as oxygen carrier [16]. In contrast, this TH metabolite 3,3′-T2 was readily formed after perfusion with rT3 (3 nmol/200 mL; estimated half-life 8.7 min) [16,17], a clear indication the of functional 5′-Dio1 activity of these livers, which also produced T3 during perfusion with T4 (26 nmol/200 mL) with an apparent conversion rate of 13% at pH 7.2. 3,5-T2 formation or its hepatic content after perfusion with the potential precursors T4 or T3 was not analyzed then, but the high stability of T3 (estimated half-life ca. 500 min) does not suggest major 3,5-T2 formation by hepatic 5′-Dio1 activity under these experimental conditions. In 1989, Horst et al. [15] discussed whether an elevated production and consequently increased serum concentration of 3,5-T2 after several days of high energy diet as reported by Burger et al. [18], would result in increased thermogenesis if this process would also occur in other major organs and not be restricted to the liver.

A second publication boost on 3,5-T2 started in 2013, when a specific nuclear T3 receptor (TR), the “long” L-TRβ1 receptor form, containing a 9 amino acid insert in its ligand binding domain, was identified in tilapia fish, which preferentially bound 3,5-T2 over T3, the classical thyromimetic hormone [19]. This insertion confers a high affinity binding (dissociation constant, K_D_: 0.2 nM) and 10^3^-fold higher sensitivity for 3,5-T2 over T3, in contrast to the “short” S-TRα1 isoform, which prefers T3 with similarly high affinity (K_D_: 0.2 nM) but a 10^3^-fold lower sensitivity as ligand. These two TR isoforms show distinctly different tissue expression patterns. Mechanistically, the L-TRβ1 form triggers similar (non)-canonical modes of rapid actions as well as transcriptional modulation of gene expression like S-TRβ1, but both isoforms might prefer interactions with distinct T3-reponse elements (TRE) and/or target genes [20]. Both, 3,5-T2 and T3 modulate TR-dependent gene expression in the liver and stimulate of myelination in tilapia brains, but their target genes and location of action differ [21]. Up to now endogenous 3,5-T2 concentrations in teleost species are unknown as are those of T3, thus comparison of their endogenous biological roles remains difficult. Whether L-TRβ1 in fish species is also involved in non-canonical signalling related to rapid THM effects has not been studied.

## 2. Direct Mitochondrial Actions of 3,5-T2

Among many other effects, TH activates mitochondrial respiration, improve hepatic and systemic glucose mechanism, and invoke anabolic and catabolic reactions of hepatic lipid turnover. Some of these TH effects have previously been ascribed to impaired oxidative phosphorylation and increased mitochondrial uncoupling similarly to the mode of action of 2,4-dinitrophenol (DNP), whose structure resembles the phenolic ring of iodothyronines. However, the TH precursor 3,5-di-iodo-L-tyrosine (DIT), which more closely mimics the structure of DNP, did not show uncoupling actions, [22,23]. DNP, a protonophore, triggers the release of calcium from mitochondrial stores, increases free intracellular calcium, and induces expression of cytochrome c oxidase-IV (COX), adenine nucleotide translocase (ANT) and uncoupling protein (UCP), effects contributing to mitochondrial uncoupling [24]. Unphysiologically high micromolar TH (T4, T3) concentrations were needed to uncouple mitochondria in vitro to a similar extent as observed for DNP. No stereospecificity for the natural L-form of TH was observed, T4 was more potent than T3, and iodide itself, in contrast to molecular iodine I2, was inactive [23]. These early studies have already revealed marked discrepancies between in vitro effects of very high TH concentrations (10^-6^ to 10^-4^ M) and in vivo actions observed after administration of physiologically low TH doses on oxygen consumption, BMR, thermogenesis and tissue-specific mitochondrial activation [22,23,25]. Circannual variations of mitochondrial TH sensitivity were also previously noted in these in vivo experiments [25].

The first hints on stimulation of mitochondrial amino acid incorporation and the induction of T3-dependent proteins interfering with oxidative phosphorylation were available since 1965 and suggested a high TH sensitivity of mitochondria [26]. UCP1 was identified as the first mitochondrial UCP, eventually purified and isolated [27] and its regulation by T3 was demonstrated by Bianco and Silva in 1987 [28]. UCP1 proved to be one major hand to exert canonical T3 action on mitochondrial thermogenesis and function in close concert with the adrenergic system. Further mitochondrial UCP were identified in BAT and later on also in WAT and muscle [29]. While the TH-sensitivity of mitochondria was previously known, 3,5-T2 gathered the attention of the researchers after Horst et al. [15] described the rapid and cycloheximide independent stimulation of oxygen consumption by 3,5-T2.

The experiments describing the influence of 3,5-T2 on the mitochondrial COX complex via a specific binding with the Va subunit further inspired researchers to investigate the effects of 3,5-T2 on the basal metabolic rate (BMR) of severely hypothyroid rats measured under the standard conditions, established at the time. The effects of an acute 3,5-T2 dose (25 µg/100 g BW) were found to be rapid within a few hours while the same, acute dose of T3 resulted in slower yet long-lasting effects for several days. The effects of T3, but not of 3,5-T2, were inhibited by the protein synthesis inhibitor Actinomycin D, suggesting a different mode of action for the two TH metabolites [30]. Lombardi et al. also showed a rapid stimulation of both cytochrome oxidizing and reducing apparatus in the respiratory chain of the rat livers with a dose of 150 µg/100 g BW 3,5-T2 via binding with the COX complex [31]. High fat diet (HFD) fed overweight rats treated daily with an injection of 25 µg/100 g BW 3,5-T2 showed a significant increase in mitochondrial fatty acid oxidation and reduction in the oxidative stress thereby reducing the pre-existing fat accumulation and hyperlipidaemia [32]. Sirtuin 1 (SIRT1), an NAD^+^^/^^−^ dependent histone deacetylase, was identified as one of the main TRβ-independent targets of 3,5-T2 and plays a key role in the rapid increase of the hepatic fatty acid oxidation and inhibition of lipogenesis [33]. 3,5-T2 is also thought to be involved in enhancing peroxisome proliferator-activated receptor (PPAR)-α as well as-δ signaling along with reprogramming lipid/glucose utilization and mitochondrial biogenesis. With the 30 days treatment of 25 µg/100 g BW 3,5-T2 in HFD-fed male Wistar rats, the protein levels of mitochondrial respiratory complexes I and II were found to be restored suggesting that mitochondria are the principle target for 3,5-T2 [34,35].

The fatty acid oxidation stimulated by a single acute dose of 3,5-T2 (150 µg/100 g BW) was also assisted by the enhanced activity of F0F1-ATP synthase in the liver of hypothyroid rats, particularly when succinate or L-palmitoyl-carnitine were provided as respiratory substrate. This effect may also be associated with altered levels of mitochondrial cardiolipin, which is required for the optimal activities of respiratory complexes and carnitine palmitoyl-transferase I (Cpt1) activity [36].

Several in vitro studies attempted to investigate the rapid actions of 3,5-T2 using pharmacologically high concentrations. The physiological relevance of these effects elicited by pharmacologically high 3,5-T2 concentrations remains unclear. A study from Rochira et al. that treated HepG2 cells with 0.01 µM–10 µM 3,5-T2, found that even 0.1 µM treatment reduced the expression of fatty acid synthase (*Fasn*) by inhibiting the proteolytic cleavage of sterol regulatory element-binding protein-1 (SREBP). However, the treatment of 3,5-T2 did not affect the mRNA levels of *SREBP-1* suggesting that the 3,5-T2 affects SREBP at translational or post-translational events [37].

## 3. Mechanisms Postulated for 3,5-T2 Actions

3,5-T2 may act as a bona fide agonist modulating the function of intracellular T3 receptors (TR), which act as ligand dependent transcription factors of T3-regulated genes. However, its binding affinity for the human TRβ1 is 40-fold lower than that of T3 [19], and much higher pharmacological 3,5-T2 concentrations are needed for its in vitro transactivation in luciferase reporter assays (median effective concentration, EC50: 3,5-T2 = 4 × 10−7 M vs. T3 = 9 × 10−12 M). Whether such high concentrations are endogenously achieved in the presence of competing T4 and T3 under physiological conditions in mammals is questionable.

Publications on direct thyromimetic effects of 3,5-T2 in vitro or in animal experimental models need to be interpreted with some caution because many commercially 3,5-T2 preparations are known to be contaminated with relevant concentrations of T3 (up to 2%). Considering the potency of T3 at its receptors, which is by more than one order of magnitude higher than that of 3,5-T2, assessment of the inherent biological activity of 3,5-T2 is difficult when no concomitant comparative control experiments are performed with corresponding T3 concentrations (as well as with iodide concentrations equimolar to those of 3,5-T2 in order to exclude unspecific effects of reactive iodide ions).

## 4. Effects of 3,5-T2 in Tissues of the HPT Axis

Chronic administration of pharmacological 3,5-T2 doses changes this scenario in rodent experimental models because 3,5-T2 rapidly and efficiently suppresses the hypothalamus-pituitary-thyroid-periphery (HPTP) axis resulting in low serum concentrations of TSH, T4 and T3 (Figure 3). Moreover, tissue T4 and T3 concentrations are also suppressed while some tissues may accumulate 3,5-T2 to sufficiently high levels. This 3,5-T2 accumulation in the tissues may alter the expression of T3-regulated genes with 3,5-T2 acting as TR ligand as shown for pituitary TSH (transcript and protein) and hepatic genes in mice [38]. 3,5-T2 also suppresses hypothalamic TRH while no such effect was observed for TRH mRNA at doses that already decreased TSH [39].

### 4.1. 3,5-T2 and the Pituitary

Apparently, in mice, the desired beneficial metabolic effects of 3,5-T2 on hepatic lipid and glucose metabolism, lipid accumulation in various organs, weight loss, etc., cannot be achieved without provoking adverse “side effects” such as TSH suppression and increased heart weight (see the Section 6).

Ball et al. reported different relative thyromimetic potencies of T3 vs. 3,5-T2 (highly pure) in their studies using in vitro pituitary models and an in vivo model of male rats made hypothyroid by PTU treatment. While both hormones (100 μg 3,5-T2/100 g BW/d and 1 μg T3/100 g BW/d, i.p. for 7 d) were almost equipotent in inducing hepatic malic enzyme transcripts, 3,5-T2 was by orders of magnitude less potent than T3 with respect to various readouts and endpoints relevant to pituitary function and regulation (TR binding and activation, GH and TSH expression) [40]. Albeit authors observed a trend for the higher affinity of 3,5-T2 to the TRβ2 isoform, they concluded that the basis for this differential thyromimetic activity of 3,5-T2 vs. T3 at different target sites studied was not related to any TR-isoform selective affinity of 3,5-T2 but probably inherent to the local cellular context of factors and mechanism involved in canonical (and non-canonical) thyromimetic action at various target structures. In 2017, da Silva Teixeira et al. observed only weak activation of TR isoforms by high concentrations of 3,5-T2 (1 µM) using in vitro liver cell models [41]. In pituitary GH3 cells, rapid increases in [Ca^2+^] were observed by remarkably low 10 nM 3,5-T2 and T3 concentrations, involving also the mediator NO in the signaling, but these effects were not studied in more detail [42].

Direct administration of 100 nM and 1 µM 3,5-T2 to isolated perfused rat pituitaries decreased TSH secretion after a TRH pulse [43]. 3,5-T2 induced Dio1 activity in a pituitary reaggregate culture model (30 nM) and GH3 cells (3 nM) and a single injection of 3,5-T2 to male euthyroid rats suppressed TSH expression and secretion after 24 h at an approximately 10-fold lower potency and shorter persistence of the effect compared to T3 administration [44]. Already (chronic) low doses of 3,5-T2 (25 µg/100 g bw/d or higher) were reported to efficiently suppress TSH and T4 secretion in various hypothyroid rat models [19,30,39,44] without exerting other metabolically relevant thyromimetic effects, an issue still controversial considering other published evidence for rapid, non-canonical stimulating effects of 3,5-T2 on mitochondrial activity, oxygen consumption, energy expenditure, and lipid metabolism. Reports also differ in their dose-response relationships between TSH and concomitant T4 suppression, probably related to experimental differences between their animal models (duration of treatment, time points of analysis, way of inducing hypothyroidism, etc.). Increasing suppression of the HPTP axis by increasing doses of 3,5-T2 was also observed in mouse and rat models (euthyroid, lean, HFD obese) [1,38,39,45].

### 4.2. 3,5-T2 Effects May Depend on the Thyroid State

Responses to 3,5-T2 depend on the TH status of the experimental animal model or the ex vivo samples studied. Impressive reproducible data were obtained in severely hypothyroid rats where both TH synthesis was blocked by PTU and peripheral activation of T4 to T3 by Dio1 (PTU sensitive) and Dio2 (PTU insensitive) was prevented by iopanoate, the inhibitor of all three Dio isoenzymes. Some authors used rats rendered hypothyroid by administration of 125-iodide 28 d before. Extent and duration of hypothyroidism differ between publications and impedes their comparability, even more so in combination with HFD administered. HFD are known to alter TH status and TH responsivity in a tissue-specific manner, as recently demonstrated for different responses of transcriptome and proteome parameters to 3,5-T2 treatment in euthyroid mice kept on regular chow vs. HFD [38,46].

In several in vitro and in vivo models, different responses (extent, kinetics, latency, and persistence) of endpoints and readouts of mitochondrial functional parameters were reported when effects of 3,5-T2 were compared with those of T3 in the absence or presence of PTU (pre-or co-) treatment. Authors interpreted those data as evidence for rapid direct action of 3,5-T2 assuming that PTU (and iopanoate) would effectively inhibit (local or systemic) 5′-deiodination of T3 to 3,5-T2. However, these results were inconsistent and rarely the effectiveness of inhibitor treatment has been documented by appropriate measurements of THM, in vivo. Moreover, a clear demonstration of enzymatic 5′-deiodination of T3 to 3,5-T2 in vitro has not been documented so far using isolated or recombinantly expressed Dio preparations or cells expressing these enzymes [2,47,48]. Human in vivo studies on kinetics and turnover of TH, using several paradigms including radioiodine-labelled TH precursors such as T4 or T3 as well as established models to determine production, turnover, and metabolic clearance rates for 3,5-T2 supported the hypothesis that 3,5-T2 is generated from T3 by deiodination in vivo outside of the thyroid gland. The thyroid gland probably does not secrete 3,5-T2, which can hardly be detected as iodinated amino acid constituent of Tg. However, most of these kinetic studies used immunoassays for quantification of serum concentrations of 3,5-T2 and it is well known that 3,5-T2 is the THM with by far the widest concentration range reported in the literature. The wide concentration range, reported for 3,5-T2, is possibly related to the notorious interference of T3 in these immunoassays, many of which have high cross-reactivities with T3 (and other THM) [49].

## 5. Hepatic Effects of 3,5-T2

### 5.1. Canonical 3,5-T2 Actions in the Liver of Rodents

In 2015, we [1] observed that a 28-day 3,5-T2 treatment (2.5 µg/g BW) in obese mice, like T3 administration, resulted in the upregulation of the classical hepatic T3 target genes like cytochrome P450 subfamily 7a (*Cyp7a*), polypeptide 1 (*Cyp7a1)*, malic enzyme (*Me1*), *Cpt1a*, and glycerol phosphate dehydrogenase 2 (*Gpd2*). It also downregulated the stearoyl-Coenzyme A desaturase 1 gene (*Scd1*), coding for a major enzyme in hepatic lipogenesis converting saturated fatty acids into monounsaturated fatty acids. Induction of *Cyp7a1* might have been responsible for improving the serum cholesterol profile and regulation of *Cpt1a*, *Me1* and *Scd1* suggested a change in hepatic lipid handling. The induction of the above genes might explain the significantly reduced hepatic triglyceride content that is observed after 28 days of 3,5-T2 treatment. This altered expression of genes involved in cholesterol and lipid metabolism by 3,5-T2 was also accompanied by a slightly increased expression of genes involved in mitochondrial biogenesis. The significant increase in the activity of citrate synthase upon 3,5-T2 and T3 treatment suggested an enhanced mitochondrial oxidative capacity.

Another classical hepatic T3 target gene is *Dio1*, which is typically upregulated by T3 to increase the hepatic TH metabolism. On 2.5 µg/g BW 3,5-T2 treatment, Dio1 transcript and activity were also found to be upregulated. We further confirmed the upregulation of *Dio1* along with the downregulation of thyroxine-binding globulin encoding gene, Serpina7 with a transcriptomic analysis [38]. The levels of Dio1 were increased and the secreted thyroxine-binding globulin was not detectable in both diet groups in the proteome analysis [46]. The modulation of these classical TH responsive genes could indicate a potential binding and activation of 3,5-T2 to the TR isoform predominating in the liver, i.e., TRβ1, thereby activating the canonical pathway of TH signaling.

After performing a comparative liver transcriptomic and proteomic analysis of HFD-induced obese mice, treated with two different doses of 3,5-T2 (0.25 and 2.5 μg/g BW), we characterized previously unknown TH-responsive genes [38,46]. The transcriptomic analysis revealed that the 3,5-T2 treatment altered the expression of genes encoding enzymes for thyroid, steroid hormone, and drug metabolism. The drug metabolizing enzymes (DME) which include Phase I and phase II enzymes and Phase III transporters, represent a defence system that protects the organism via the metabolism and elimination of xenobiotics and endobiotics [50]. The proteomic analysis of 3,5-T2 treatment on both HFD-fed vs. ND-fed mice revealed increased levels of isocitrate dehydrogenase 3 subunits Idh3a and Idh3g. The transcriptional activation of *Idh3a* is an effect of classical canonical genomic signaling and has been previously demonstrated in mice [50,51,52].

### 5.2. Noncanonical 3,5-T2 Actions in the Liver of Rodents

The PI3K-Akt/PKB-mTOR cascade is crucial for cell metabolism, growth, proliferation, and survival. Dysregulation of the PI3K/Akt pathway in hepatocytes has been associated with metabolic dysfunctions like obesity and non-alcoholic fatty liver disease (NAFLD) [53]. Previous studies have shown the rapid activation of the PI3K-Akt/PKB-mTOR cascade via the noncanonical action of TRβ [54]. Activation of PI3K further stimulates transcription of hypoxia-inducible factor (Hif)-1α [55]. Hönes et al. demonstrated that the noncanonical TR signaling significantly contributed to the regulation of energy homeostasis and suggested a convergence of insulin signalling and noncanonical TRβ signalling at the PI3K pathway [56].

With the proteomic analysis, we revealed a large number of genes that were differentially expressed at the protein levels with 3,5-T2 treatment [46]. Proteome alterations are fixed due to any consequent changes in hepatocellular proteostasis, while alterations in the transcriptome might not be detected due to the resetting of early-induced transcriptional responses. The proteome alterations may be attributed to post-transcriptional regulatory mechanisms, that are either modulated by the cytoplasmic THR dependent non-canonical signaling of 3,5-T2 or via THR-independent modulation [46].

### 5.3. Lipid Metabolism

TH are long known for reducing blood concentrations of cholesterol [57] and both T3 and 3,5-T2 exerted such action in mice fed a western diet, irrespective of the presence or absence of the Ldlr or induction of the hepatic scavenging receptor B1 [45], which typically increases after administration of synthetic thyromimetics. 3,5-T2 administration to HFD rats (i.p. injections of 25 μg/100 g b.w.-d for 4 w) or HFD mice (2.5 µg/g b.w.-d, i.p. for 4 w) reverted hepatic triacylglycerol accumulation and apart from enhanced Cpt1a activity [32], several components involved in triacylglycerol degradation were activated [43]. However, intrahepatic triglyceride content was unaltered in the clinical trial using the synthetic 3,5-T2 analogue TRC150094 [58].

3,5-T2 treatment (0.25 mg/kg-d) for 10 or 30 days of adult male Sprague Dawley rats (kept on a safflower-oil based high-fat diet) revealed no change in the expression of T3 responsive hepatic genes involved in the regulation of lipid oxidation, de novo lipogenesis, glycolysis or gluconeogenesis. Authors concluded that 3,5-T2 treatment may exert modest benefit for hepatic but neither for whole-body insulin sensitivity nor for improvement of NAFLD and thus its clinical utility might be very limited [59]. The team could not reproduce several previous observations on antisteatotic effects of 3,5-T2 using male Sprague-Dawley rats in their steatosis-prevention experiment.

### 5.4. Thyromimetic Effects of 3,5-T2 on Glucose Metabolism

Da Silva Teixeira et al. [41] treated male C57BL/6 mice kept on HFD for 8 weeks with a T3 dose (0.75 mg/kg/d) (known to induce hyperthyroidism), two doses of 3,5-T2 (1.25 and 12.5 mg/kg/d) and its structural isomer 3,3′-T2 (1.25 mg/kg/d). Effects of THM on metabolic and endocrine endpoints analyzed differed with respect to hormone and dose studied. T3 acted as expected for the thyromimetic ligand but 3,5-T2 also suppressed the HPTP axis, increased body temperature, food intake, heart weight, and lean body mass, while decreasing fat mass and body weight. At the high dose, 3,5-T2 acted thyrotoxic like T3, while the low dose suppressed TSH and increased heart weight but did not show the metabolic effects typical for a hyperthyroid state. The high dose of 3,5-T2 reduced fasting blood glucose and insulin concentration while glucose tolerance was not changed, a pattern different from that of T3 (and 3,3′-T2) responses. Alterations exerted by 3,3′-T2 treatment were distinct from those of T3 and 3,5-T2 as only an increased food intake was observed with trends to the increased body and liver fat and partially suppressed TSH. Treatment with T3 and both doses of 3,5-T2 but not 3,3′-T2 decreased hepatic Glut2 expression. The 3,5-T2 treatment did not enhance the markers of glucose uptake in muscle and WAT or those of insulin sensitivity in the liver. The authors suggested that 3,5-T2 reduces blood glucose by decreasing the expression of the bidirectionally active Glut2 as well as hepatic glucose output. As 3,5-T2 enhanced expression of Ucp1 and Ucp3 transcripts in WAT and muscle, these tissues might contribute to blood glucose reduction while no activation of BAT was detected. Unfortunately, sources and purities of 3,5-T2 and 3,3′-T2 used in this comparative in vivo study are not given and the fact that authors observed several “thyromimetic” effects for 3,3′-T2, typically not described in published literature, raises the concern that the 3,3′-T2 preparation used in this study might have been significantly contaminated by T3.

### 5.5. Hepatic Accumulation, Selective Uptake and Inefficient Hepatic Elimination of 3,5-T2

A significant dose-dependent increase in the serum and hepatic concentrations of 3,5-T2 after a 14-day treatment of 3,5-T2 was observed in diet-induced obese mice (Figure 4) [1]. The unexpected rise in hepatic concentrations may either be due to an increased uptake and/or reduced elimination of the TH metabolite and warrants further studies [49]. The classical transporters for thyroid hormones such as Mct8, Mct10, Oatp, and Lats, however, show a low transport activity for 3,5-T2 compared to T4, T3, or 3,3′-T2 [60,61]. However, the anti-hyperlipidemic effects of 3,5-T2 and its accumulation in the liver, as observed by our group are similar to the features of the TR-selective ligand eprotirome [1,2]. Eprotirome is a liver-specific drug with a high affinity towards TRβ and is shown to halt the hyperlipidemic constellation in human volunteers [62]. Previous studies reporting a highly effective induction of hepatic *Me1* gene expression by 3,5-T2 compared to the suppression of TSH [40] may be explained by the accumulation of 3,5-T2 in the liver. Consequently, the effects of 3,5-T2 may not only depend on differential binding to the TR-isoforms but also depend on the high local availability of the TH metabolite in the liver [1].

### 5.6. Implications of High 3,5-T2 in Context of Low T3, T4, and Suppressed TSH for Hepatic Lipid Metabolism

The decreased serum and hepatic concentrations of T4, T3, and rT3 upon a high dose of 3,5-T2 and a downregulated gene expression of TSH-releasing hormone receptor (*Trhr*), *Dio2*, and *Tshb* indicate a physiological suppression of the HPTP axis. However, the observed upregulation of Dio1 in the liver on 3,5-T2 treatment is similar to the upregulation of Dio1, typically observed upon TH-induced hyperthyroidism [1].

Thus, the enhanced hepatic Dio1 activity along with altered serum TH concentrations and pituitary transcripts on 3,5-T2 treatment are indicative of a local hyperthyroidism accompanied by accumulation of hepatic 3,5-T2, which apparently is not deiodinated by high Dio1 activity. Additionally, our transcriptome analysis showed increased expression of genes encoding sulfotransferase and UDP-glucuronosyltransferase (*Ugtp*) in mice treated with 2.5 µg/g BW/d 3,5-T2 for 28 days compared to untreated mice [38]. In primary hepatocytes, the 3,5-T2 treatment concentration-dependently increased the expression of *Sult1a1*, potentially causing a subsequent inactivation of 3,5-T2 as observed for T3 [63]. Isoforms of Sult and Ugtp have been previously shown to play an important role in iodothyronine conjugation and enterohepatic recycling. These enzymes are also involved in maintaining T3 concentrations for TR binding and activation [64,65,66]. Silvestri et al. [67] observed some differences between the protein levels of Sult1a1 between ND-fed vs. HFD-fed rats treated with 0.25 µg/g BW/d 3,5-T2. However, no changes were observed in the protein levels of Sult1a1 in the liver of 0.25 µg/g BW/d 3,5-T2 vs. sham treated HFD-fed rats.

Taken together, the increased expression of hepatic Dio1 and sulfotransferases from the 3,5-T2 treatment may affect and steer the local TH concentrations, which needs further attention and systematic studies.

### 5.7. Hepatic 3,5-T2 Effects Differ between Lean and HFD Obese Mice

Considerable efforts were taken to describe the beneficial effects of 3,5-T2 on mitochondrial function, lipid mobilization, and fatty acid oxidation in obese or hypothyroid rodents [33,36,68,69]. However, our group attempted to compare the effects of a 28-day 3,5-T2 treatment in lean and diet-induced obese euthyroid mice by employing a transcriptome and proteome analysis [38,46]. While the genes encoding the drug metabolising enzymes in the liver were similarly affected in both lean and obese mice, 12 genes were found to be differentially expressed in the obese mice. An increased oxidative stress and/or hepatic inflammation as indicated by activation of NRF-2 targets of ROS defence and a subsequent crosstalk between pro-inflammatory factors and biotransformation enzymes in the fatty liver may have modified expression of genes encoding drug metabolising enzymes (DME) [70]. This, in turn, might have caused a differential expression of genes and also be contributing to the elimination of exogenously added 3,5-T2. Another possible explanation for the differential gene expression in lean and obese mice may be due to the overlap of gene expression profiles that are altered by HFD with those modulated by 3,5-T2. The 3,5-T2 treatment in such cases may not act additively or synergistically with HFD.

Further differences were found after a 4-week 3,5-T2 treatment in the expression of the genes relevant for farnesyl, cholesterol, and bile acid biosynthesis. The expression of this gene set was only influenced in obese mice, suggesting a novel role of 3,5-T2 in the steroid metabolism of obese mice. 3,5-T2 treatment-induced gene expression of enzymes involved in the biosynthesis of zymosterol, a precursor of cholesterol [71]. An increased secretion of biliary cholesterol and increased hepatic cholesterol synthesis upon 3,5-T2 treatment may play an important role in the cholesterol elimination thereby providing 3,5-T2 its antisteatotic ability. Differential expression of glutathione-related redox-reactions and detoxification pathways as well as of components of triacylglycerol degradation indicate a remarkable impact of 3,5-T2 on hepatic intermediary metabolism in context of LXR/RXR and FXR/RXR activation in HFD-obese mice.

## 6. 3,5-T2 and the Heart

Doses of 3,5-T2 required to exert ‘beneficial’ metabolic effects on liver (lipid) metabolism, energy expenditure, glucose metabolism and effects on muscle and adipose tissue, resulting in adverse cardiac effects over time, typical for potent thyromimetic compounds not selectively targeting the liver, adipose tissue or brain [1,39,58,72,73].

Bovine heart mitochondria are targets of 3,5-T2 action [74,75]. However, the exact mechanisms and proteins involved are still under investigation. One of the first mitochondrial targets proposed to mediate rapid direct 3,5-T2 effects was the Va subunit of the mitochondrial cytochrome C oxidase (COX), a key element of the respiratory chain. In vitro, 3,5-T2 treatment at high concentrations >10^−7^ M stimulated isolated cytochrome c oxidase from bovine heart mitochondria at pH 7.4, while T3 did not show such an effect [74].

Mechanisms involved in mitochondrial targeting by 3,5-T2 (direct or indirect), effects of 3,5-T2 due to changes in Ca^2+^ signaling, mitochondrial ion currents, mitochondrial charge or activity, and involvement of mitochondrial p43-TRα form [76] and ADN-translocase remain under discussion [55]. Obviously, the mitochondrial response to acute or chronic 3,5-T2 exposure differs between mitochondria of the hepatic, brain, and skeletal or cardiac muscle origin, and previous in vivo TH status of the animal impacts on the outcome of in vitro exposure to 3,5-T2 [77]. Whether the mitochondrial p43-TRα form is also involved in non-canonical mechanisms of THM action or modulation of THM-TR-regulated transcription of mitochondrial DNA is unknown.

Sacripanti et al. exposed the rat cardiomyoblast cell line H9c2t for 4 h to 0.1–10 μM 3,5-T2 concentrations and studied glucose uptake [78]. They observed cellular uptake of 3,5-T2 by LC-MS analysis and increased glucose uptake by 24 or 35% in cells exposed to 0.1 or 1.0 μM 3,5-T2, concentrations which did not affect cell viability. T3 did not cause such changes and T4 (1 and 10 µM) even reduced glucose uptake. Effects reported for 3,5-T2 perfused isolated rat hearts required 10 µM 3,5-T2 concentrations questioning their relevance [78].

In hypothyroid cold exposed rats 3,5-T2 treatment (2.5–10 µg/100 g BW) stimulated the COX complex in the heart and other tissues (liver, BAT, skeletal muscle) similar to T3 (2.5 µg/100 g BW/d), enabled survival in a cold environment over 3 weeks, but did not increase heart weight as observed for T3 treatment [79]. This difference suggested another, potentially beneficial, mode of action for 3,5-T2 under avoidance of unwanted cardiac side effects regularly associated with T3 treatment. However, the expression of COX transcripts and protein also depends on TH status in a tissue-specific manner [80].

Chronic administration (daily, s.c. for 90d) of 3,5-T2 (50 µg/100 g BW/d) with or without T4 (1 µg/100 g BW/d) to adult male rats increased heart/body weight ratios but protected against ischaemia–reperfusion injury as shown by reduced infarct size concomitant to an improved autonomous heart control [81]. 3,5-T2 enhanced systolic arterial and mean blood pressure as well as baroreflex sensitivity in both groups which had either suppressed (3,5-T2 alone) or regular T4 and T3 serum concentrations (3,5-T2 co-treated with T4). The authors discussed various possibilities of underlying mechanisms and indicated the potential translational relevance of these preclinical observations. It remains to be studied whether chronic 3,5-T2 treatment is acting via mitochondrial effects, canonical TR mechanisms or similar to non-canonical TRα mediated action reported for T3, which rapidly leads to endothelium-dependent vasodilation via PI3K and eNOS activation [82]. 3,5-T2 action has not been studied in experimental (mouse) models of left ventricular pressure overload, cardiac hypertrophy and congestive heart failure where TH deprivation was protective [83]. The repetitive demonstration of heart weight increases in mice after chronic 3,5-T2 administration [2] which would be required for intended sustainable anti-steatotic effects raised concerns about such a strategy.

In a cold-adapted hypothyroid zebrafish model 3,5-T2 supplementation restored resting heart rate similarly to T3, accompanied by normalization of maximal SERCA activity, indicative of a canonical mechanism of 3,5-T2 action. While T3 treatment resulted in control level maximal heart rate, this was not achieved by 3,5-T2 treatment [84]. Hypothyroidism was induced by treatment with propylthiouracil and iopanoate over a 3-week acclimation period to 18 or 28 °C. However, the response to 3,5-T2 treatment depended on the environmental temperature, and in this zebrafish model T3 and 3,5-T2 effects on swim performance and some metabolic aspects differed in direction and extent. Evidence was also presented for muscular transcriptional effects of 3,5-T2 and restoration of metabolic rate, while basal metabolic rate and hepatic expression of TH responsive genes were unaltered by 3,5-T2 [85]. Muscular 3,5-T2 contents achieved in control and hypothyroid fish were lower than those obtained by T3 treatment as determined by LC-MS/MS analysis. Unfortunately, concentrations used for TH treatment remained undefined. Whether the short S-TRβ1 isoform found in tilapia might be involved in 3,5-T2 effects in the zebrafish model has not been studied [86].

## 7. Dose Issues of 3,5-T2

Several studies described the beneficial antisteatotic and metabolic effects of 3,5-T2 especially on the liver or muscle tissues without reporting any adverse side effects usually observed on T3 administration [33,87]. However, many of the studies that used rat models do not inform on the concentrations of 3,5-T2 in circulation or in various target tissues upon acute and chronic administration of 3,5-T2 [88,89]. The abovementioned studies, while describing alterations in the hepatic gene expression on 3,5-T2 treatment, also do not describe any suppression of the HPTP axis.

We compared the effects of two doses of 3,5-T2 (0.25 and 2.5 µg/g BW) with T3 treatment on the adiposity and regulation of the HPTP axis in diet-induced obese male mice for 14 days followed by a longer 28-day dose regimen (Table 1) [1]. A dose-dependent increase in both serum concentrations and hepatic concentrations of 3,5-T2 was observed on 3,5-T2 treatment whereas a T3 treatment did not affect the serum T3 concentrations but resulted in increased concentrations of T3 in the hepatic tissue. Both 3,5-T2 and T3 doses reduced the serum T4 concentrations indicating a physiological suppression of the HPTP axis through the negative feedback mechanisms. The low dose of 3,5-T2 did not alter the expression of most genes related to the HPTP axis regulation that were down-regulated by T3 treatment. However, the higher dose of 3,5-T2 effectively suppressed the same genes. The classical hepatic T3 targets, like deiodinase enzyme type I (*Dio1*), malic enzyme (*Me1*) or *Cpt1a*, were only up regulated with the higher dose of 3,5-T2 and the lower dose was found ineffective. Dose dependent differences were also observed for the genes involved in mitochondrial biogenesis. The metabolically active dose of 3,5-T2 was found to be effective after 14 days in its T3-mimicking actions. The observed suppression of the HPTP axis and a local hepatic hyperthyroidism on 3,5-T2 treatment in mice [1] is in line with the observations of Padron et al., [39] describing a dose dependent suppression of serum TSH, T4, and T3, post 3,5-T2 treatment (25 µg, 50 µg and 75 µg/100 g BW/d) in male Wistar rats (Figure 5). Only at the highest 3,5-T2 dose, did Padron et al. [39] observe an increase in the relative heart mass of the 3,5-T2 treated male Wistar rats. Goldberg et al. [45], however, observed relative increases in heart weight of the 3,5-T2 (1.25 mg/kg BW and 12.5 mg/kg BW) treated mice kept on a western diet and suspected that the chosen 3,5-T2 dose may potentially be responsible for the cardiac toxicity.

3,5-T2 is known to bind to the TR with a lower affinity compared to T3. However, similar effects of the high 3,5-T2 dose and T3 indicate at least a partial action of 3,5-T2 via the (canonical) nuclear pathways. Therefore, 3,5-T2 at higher concentrations than T3 may act as a potential ligand for at least some TR isoforms. These observations highlight the need for studies to further characterize both beneficial and adverse effects of 3,5-T2 in different species and to consider the differences in hepatic and pituitary 3,5-T2 uptake, metabolism, action, and elimination.

## 8. Limitations, Deficits, Problem Areas and Challenges in 3,5-T2 Research

One of the major deficits in 3,5-T2 research is the lack of monitoring of the alterations of systemic and local THM concentrations when the thyromimetically active ‘hot’ 3,5-T2 is administered acutely or chronically in high experimental pharmacological or assumed physiological doses. These interventions alter the HPTP axis and its feedback regulation, which dynamically reacts by either adaptation or compensation in this perturbation scenario as illustrated by decreased TSH in several models. Rarely, dose-response relationships have been studied or documented, and marginal increases or decreases e.g., compared to the “standard 25 or 50 µg/kg bw/d doses” in rats, lead to loss of “beneficial” effects in one target tissue (e.g., liver) or adverse side effects in another organ (e.g., heart). The majority of studies were performed in (severely) hypothyroid rats but effects in euthyroid controls are rarely compared. Similarly, frequently 3,5-T2 (in comparison to T3) is administered to hypothyroid rodents on HFD while no hypothyroid and euthyroid lean controls are comparably analyzed.

Many in vitro studies used unrealistically high, up to mM (!) 3,5-T2 (and T3) concentrations in cell culture studies or for incubations of isolated mitochondria prepared from hypothyroid animals. Such publications were not reviewed here in detail. Such high µM THM concentrations in vitro result in rapid mitochondrial swelling and uncoupling [90], processes irrelevant to physiological TH action or studies unraveling mitochondrial effects of THM. Considering the high load of the redox-reactive iodine introduced into the in vitro test models by mM or high µM 3,5-T2 (typically no concomitant iodide controls are analyzed) rapid ‘non-canonical’ iodide effects e.g., on mitochondrial functions, might be misinterpreted.

During the last 5 years, an unhealthy ratio between published original contributions and reviews on 3,5-T2 issues evolved trending towards one review per two original reports. Furthermore, critical discussions of limits and interpretation of published in vitro and in vivo studies are avoided and gave way to ‘encyclopaedic or superficial overviews’. This concerning development is obviously boosted by (too) many new open access and commercially driven journals popping up recently and problems in maintaining a scientifically adequate and rigid peer review process.

An issue inherent to translation-oriented molecular life sciences is the use of (highly) artificial cellular models to gain mechanistic information. In the case of TH research, in vitro cellular conditions are usually chosen, where endogenous binding and distributor proteins for the high affinity binding of THM [91,92] are either absent or omitted on purpose. Typically, in vivo, less than 0.1% of TH circulate in free, i.e., bioavailable form, which is an almost unique situation among the low molecular weight hormonal ligands for their (nuclear) receptors that act as ligand regulated transcription factors in close crosstalk with other canonical and non-canonical signaling cascades. The absence of the endogenous (serum and cellular) TH binding proteins in cellular or subcellular in vitro models may result in pharmacologically high local TH concentrations prone to distorted TH metabolism and resulting metabolite patterns differing from in vivo conditions with respect to their compartmentalization and bioavailability. In vivo 3,5-T2 concentrations are very low compared to those reported for T3 or the even higher T4 in circulation and peripheral tissues (as far as information is currently available on this issue), but might be different in the brain, specific organ compartments, or ‘fatty’ peripheral tissues. Nevertheless, it appears unrealistic to expect 3,5-T2 concentrations to exceed those of T3, which are already very low compared to those of the prohormone T4. Obviously, with decreasing iodination grade of THM their lipophilicity and thus also binding affinity to TH distribution proteins is diminished and their free fraction may increase, but not to an extent that free fractions of 3,5-T2, T3 and T4 will reach the same order of magnitude.

Concerning biosynthesis and in vivo origin of 3,5-T2 it appears unrealistic (chemically and biochemically) to assume that 3,5-T2 might directly originate from T4 bypassing T3 as an intermediate. Despite many attempts, there is no evidence in vitro yet for the direct deiodination of T3 to 3,5-T2 by any of the three Dio isoenzymes. This would be a theoretically plausible metabolic pathway suggested by several in vivo animal experiments using T3 vs. 3,5-T2 and inhibition of Dios by either PTU or iopanoate or both. Recent studies in two different human cohorts, treated with T3 at different doses, time points and TH status, provided no evidence for the in vivo formation of 3,5-T2 under conditions chosen and with a 3,5-T2 immunoassay used for serum analysis; unfortunately, no parallel LCMS measurements were possible due to lack of sample volume and funding [2]. The latter observation was unexpected considering estimates of T3 → 3,5-T2 conversion rates calculated from previous in vivo kinetic analyses of TH metabolism, production, metabolic clearance, and turnover rates using radioisotope labeled THM and complex modeling of kinetic data or THM measurements with 3,5-T2 immunoassays. However, for none of the THM so far analyzed in serum by ‘specific’ immunoassays’, such a high concentration range and difference between assays used was reported than for 3,5-T2 [2]. This observation raised concerns both with respect to the analytics and potential physiological functions of 3,5-T2, which may not be a typical systemically circulating TH (metabolite), but possibly an intracellular TH intermediate or signaling compound. Attempts to determine serum 3,5-T2 concentrations by LC-MS methods require extensive pre-analytical sample extraction and workup as well as particular LC-MS procedures. These attempts revealed either a much lower concentration than those reported by immunoassays if an LC-MS/MS/MS approach was chosen [93], or low but overestimated values [94] or even failed to detect 3,5-T2 in serum [95].

## 9. Clinical Potential

In the last few years, various attempts are being undertaken to develop T3 analogue-based drugs for widespread metabolic diseases, but systemic exposure to high T3-(analogue) doses must be avoided. This can be achieved by designing liver-targeting T3-prodrugs [73,96,97], which selectively “clean” hepatocyte lipid stores, and quite promising candidates have been or are currently in phase 2 and 3 clinical studies (ClinicalTrials.gov Identifier: NCT03900429, NCT01410383, NCT4173065). Even liver-targeting DNP-prodrugs are under investigation for their therapeutic potential in diabetes and steatosis treatment [72].

TRC150094, a synthetic analogue of 3,5-T2, has also been reported to have positive effects on lipid metabolism, and prevention of hepatosteatosis at comparable doses. TRC150094 has not been shown to undergo any metabolic modifications like T3-antagonistic thyronamines or its thyroacetic acid oxidation products. This may possibly be relevant in explaining the accumulation of 3,5-T2 in the hepatic tissue [98,99].

Early studies with this analogue showed potent, beneficial metabolic effects in experimental animal models without any potentially adverse cardiac and CNS side effects [100,101,102]. However, two randomized Phase I clinical trials (clinicaltrials.gov NCT01408667; Clinical Trial Registration: EUDRA CT: 2009-014941-10 (SAD)) using a synthetic mimetic of 3,5-T2, TRC150094, failed to show similar results. One of the studies failed to meet clinical expectations but was safe in a 4-week trial at a dose of 50 mg in overweight and obese subjects at an increased cardiometabolic risk [58]. The second study, however, reported a trend in the improvement of the triglyceride profile and a significant reduction in Apo B in obese and elderly subjects at a dose of 50 mg/day [103]. Reported species differences in 3,5-T2 production, metabolism, and (physiological) function complicate the assessment of its relevance and the development of 3,5-T2 mimetics such as TRC150094 as anti-steatotic drugs or agents beneficial for disorders of lipid metabolism. So far, convincing evidence has not been presented that 3,5-T2 or its synthetic analogues might be used in clinical practice for the prevention or treatment of obesity and related metabolic conditions [49] even though impressive preclinical data has been presented on remarkable anti-steatosis effects of 3,5-T2 in some rodent models [1,39,67]. Unprescribed use of 3,5-T2 obtained from the internet or other sources for purposes of slimming or body building is strongly discouraged.

## Figures and Tables

**Figure 1 metabolites-12-00582-f001:**
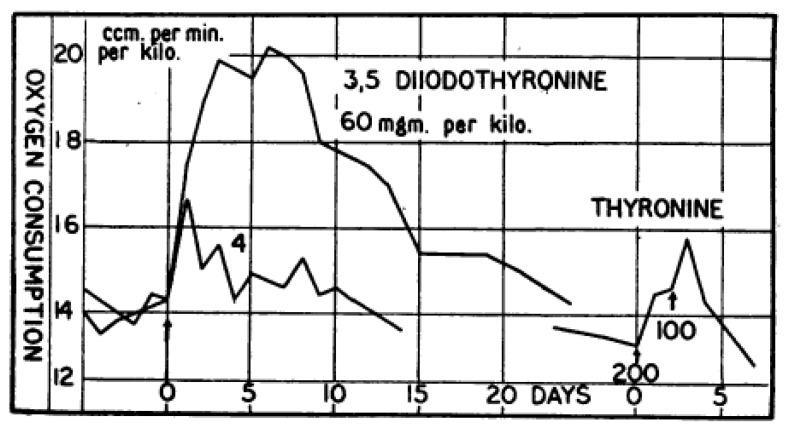
Effect of 3,5-T2 injection (s.c.) on Oxygen Consumption of Rats: Effect of 3,5-T2 injection (s.c.) on oxygen consumption (ccm/min/kg, normalized to kg body weight) of euthyroid adult male Wistar rats. Compared to T4 (not shown) 15 times higher 3,5-T2 doses elicited the same response, an observation similar to the requirement of 40-fold higher 3,5-T2 concentration compared to T4 needed to induce tadpole metamorphosis (see Gaddum 1927). Thyronine (T0) was inactive in rats after two injections of 200 and 100 mg/kg (lower right), as were diiodotyrosine (DIT) and thyroxamine (T4-Amine) (not shown). Reprinted with permission from Ref. [7], © 1930, John Wiley and Sons.

**Figure 2 metabolites-12-00582-f002:**
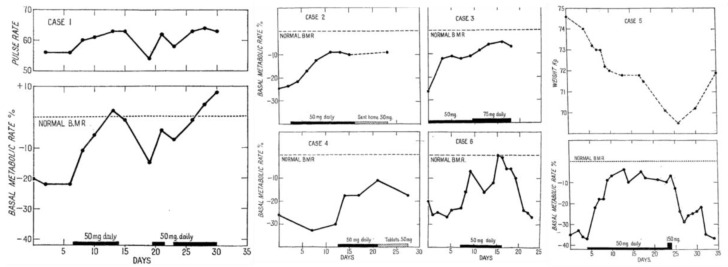
Changes in pulse-rate, basal metabolic rate, and body weight under oral treatment with 3,5-diiodothyronine (50 mg/d) of six hypothyroid individuals: Changes in pulse-rate, BMR and body weight under oral treatment with 3,5-T2 (50 mg/d) of six hypothyroid individuals. Case 1: Changes in pulse-rate and normalization of BMR under oral 3,5-T2 treatment (50 mg/d). Cases 2, 3, 4, 6: Restoration of BMR under oral 3,5-T2 treatment (initial dose: 50 mg/d). After initial daily treatment, case 2 was sent home for a week on an oral dose of 50 mg/d and then readmitted. Case 3 treated for 9 days with the initial dose of 50 mg/d and then the dose was increased to 75 mg/d for further seven days without any toxic symptoms. Case 4 received the initial daily oral 50 mg dose and after 10 days 3,5-T2 was administered as a pill on lactose base (25 mg) twice daily after food. Case 6 received the oral 50 mg/d dose restoring the BMR. Within four days of the drug being discontinued, her weight began to increase again and the pulse rate to fall (not shown). Case 5: Weight loss and restoration of regular BMR after 50 mg/d dose. The periods of oral 3,5-T2 treatment are indicated by black bars on the baseline, the switch to tablets in Case 4, and the ambulatory treatment in case 2 by shaded bars. Reprinted from The Lancet, Ref. [8] © 1933, with permission from Elsevier.

**Figure 3 metabolites-12-00582-f003:**
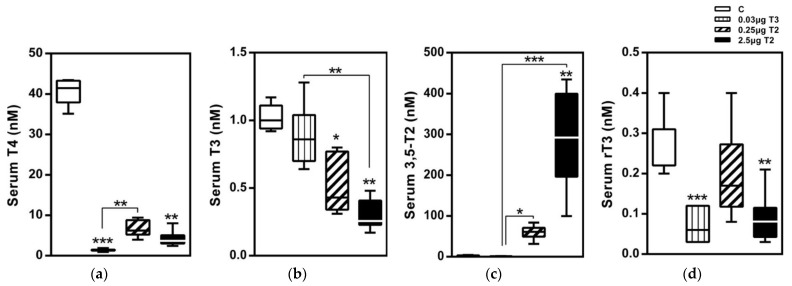
Serum TH profiles after a 14-day treatment with 3,5-T2 as measured by LC-MS/MS: Serum TH concentrations of HDF mice injected (i.p.) with either T3 (0.03 µg/g body weight) or 3,5-T2 (0.25 or 2.5 µg/g body weight) once daily at onset of dark cycle for 14 days to validate effects of 3,5-T2 in comparison with T3 (*n* = 7–8). (**a**) serum T4, (**b**) serum T3, (**c**) serum 3,5-T2, and (**d**) Serum rT3 *, *p* < 0.05; **, *p* < 0.01; ***, *p* < 0.001 vs. control or as indicated (*n* = 6–8). Reprinted with permission from Ref. [1], © 2015, Endocrinology.

**Figure 4 metabolites-12-00582-f004:**
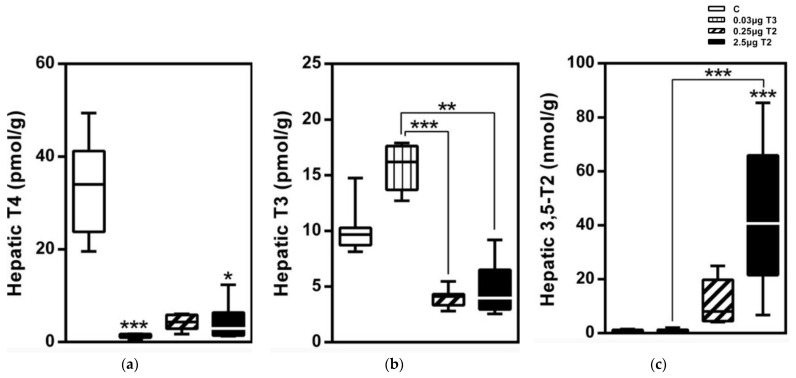
Hepatic TH profiles after a 14-day treatment with 3,5-T2 as measured by LC-MS/MS: Hepatic TH concentrations of HFD mice injected (i.p.) with either T3 (0.03 µg/g body weight) or 3,5-T2 (0.25 or 2.5 µg/g body weight) once daily at onset of dark cycle for 14 days to validate effects of 3,5-T2 in comparison with T3 (*n* = 7–8). (**a**) serum T4, (**b**) serum T3, and (**c**) serum 3,5-T2 *, *p* < 0.05; **, *p* < 0.01; ***, *p* < 0.001 vs. control or as indicated (*n* = 6–8). Reprinted with permission from Ref. [1], © 2015, Endocrinology.

**Figure 5 metabolites-12-00582-f005:**
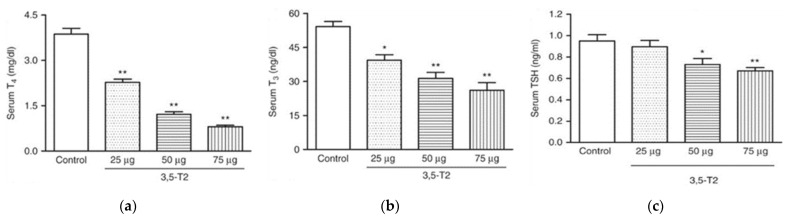
TH profile serum concentrations in adult male Wistar rats treated (or not treated) with 3,5-T2 for 90 days at daily s.c. doses of 25, 50, and 75 µg/100 g BW. Serum TSH, T4, and T3 concentrations were dose dependently suppressed by 3,5-T2 treatment. (**a**) serum total T4 (13–18 rats per group); (**b**) serum total T3 (13–18 rats per group); (**c**) serum TSH (11–14 rats per group). * *p* < 0.01 vs. control group; ** *p* < 0.001 vs. control group. Reprinted with permission from Ref. [39], © 2014, The authors.

**Table 1 metabolites-12-00582-t001:** Thyroid Hormone and Lipid Status after 3,5-T2 Treatment. Reprinted with permission from Ref. [1], © 2015, Endocrinology.

1	C	3,5-T2
Hepatic CS activity (µmol·min^−1^·min^−1^)	700 ± 30.0	1024 ± 35.7 **
Hepatic D1 activity (pmol·mg^−1^·min^−1^)	150 ± 52	380 ± 93 ***
Serum triglycerides (µg/mL)	80 ± 7.3	97 ± 12.1
Free fatty acids (mM)	0.65 ± 0.04	0.68 ± 0.04
Total cholesterol (mg/dL)	181 ± 7.1	104 ± 8.0 ***
Liver triglycerides (µg/g protein)	1.18 ± 0.39	0.73 ± 0.13 *

Data are expressed as mean ± SEM *, *p* < 0.05; **, *p* < 0.01; *** *p* < 0.001; (*n =* 6–8).

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
