# Peer review of "3,5-T2-an Endogenous Thyroid Hormone Metabolite as Promising Lead Substance in Anti-Steatotic Drug Development?"

_metabolites, 2022, doi:10.3390/metabo12070582_

Round 1

Reviewer 1 Report

It is a quite interesting analysis. Adequate details about the topic are given. I encourage the authors to add a small paragraph about TH metabolism so as non experts in the field to be able to follow this detailed review. 

Author Response

Dear Editors and Reviewers!

First and foremost, we wish to express our heartfelt thanks to the reviewers of our manuscript for their valuable and positive feedback. We sincerely appreciate the time and thoughts put forth in this manuscript by both the reviewers and the editors of the journal.

In the following section, we would like to take the opportunity to address the reviewers' comments and suggestions. Our comments are in italics blue for a clear distinction.

Reviewer1:

"It is a quite interesting analysis. Adequate details about the topic are given. I encourage the authors to add a small paragraph about TH metabolism so as non experts in the field to be able to follow this detailed review."

Please see the newly added paragraph 1, which introduces thyroid hormones and their metabolism to the scientific community allowing a deeper understanding of the subject.

Reviewer 2 Report

The authors present a narrative review of the effects of 3,5-T2 on hepatic lipid and glucose metabolism on the (non)canonical mechanisms of action, in order to fill the summary literature gap on this type of approach in relation to weight control and related metabolic issues (fatty liver etc).

Authors should specify that this is a narrative review, at least in the abstract.

It would be helpful for the authors to outline a minimum of the methodology used to search the literature databases on the topic. Where does the amount of information reported originate, and what search strategy was adopted? were all study settings taken? all populations? animal and mouse models together? This needs to be specified because it would give order to the stream of consciousness of the narrative review.

Line 420. Paragraph on the effects of thyromimetics on glucose metabolism. The authors here should dwell more on how the preservation of muscle mass can improve thyroid activity (10.3389/fendo.2020.565065), and therefore the effect of thyromimetic should be modulated based on the target patient population under consideration.

Author Response

Dear Editors and Reviewers!

First and foremost, we wish to express our heartfelt thanks to the reviewers of our manuscript for their valuable and positive feedback. We sincerely appreciate the time and thoughts put forth in this manuscript by both the reviewers and the editors of the journal.

In the following section, we would like to take the opportunity to address the reviewers' comments and suggestions. Our comments are in italics blue for a clear distinction.

Reviewer 2:

The authors present a narrative review of the effects of 3,5-T2 on hepatic lipid and glucose metabolism on the (non)canonical mechanisms of action, in order to fill the summary literature gap on this type of approach in relation to weight control and related metabolic issues (fatty liver etc).

Authors should specify that this is a narrative review, at least in the abstract.

Please see paragraph 1, where we describe the narrative nature of the review.

It would be helpful for the authors to outline a minimum of the methodology used to search the literature databases on the topic. Where does the amount of information reported originate, and what search strategy was adopted? were all study settings taken? all populations? animal and mouse models together? This needs to be specified because it would give order to the stream of consciousness of the narrative review.

For the pubmed-based primary literature search, we included specific keywords & phrases (e.g., 3,5-T2 & liver, 3,5-T2 and heart, the discovery of 3,5-T2). We then narrowed the results with a pre-defined scope of addressing the impact of 3,5-T2 on particular organs (liver, heart and the main glands involved in the feedback axis (i.e. thyroid and pituitary) in rodents and excluded the results that did not meet these criteria. Various combinations of these search terms in pubmed revealed around 200 hits, many of which were reviews including 3,5-T2 among other thyroid hormone metabolites and synthetic analogues. To address the reviewer’s suggestion in the manuscript, we have added a line at the end of paragraph 1 that defines the scope of the review and the key search-terms. We hope that this statement outlines the review and clarifies it further. Because the endogenous thyroid hormone metabolite 3,5-T2 has not been administered to humans apart from two controlled trials with the 3,5-T2 analogue TRC150094, data published on other novel synthetic T3 agonists have not been addressed. Only these two studies on effects of the synthetic 3,5-T2 analogue TRC150094 in humans (refs. #102, #103) were discussed.

Line 420. Paragraph on the effects of thyromimetics on glucose metabolism. The authors here should dwell more on how the preservation of muscle mass can improve thyroid activity (10.3389/fendo.2020.565065), and therefore the effect of thyromimetic should be modulated based on the target patient population under consideration.

The focus of our review is on the role of 3,5-T2 on liver steatosis with special consideration of its cardiac site effects and impact on the hypothalamus – pituitary – thyroid – periphery (HPTP) feedback axis. Potential further 3,5-T2 effects on adipose tissues, muscle and brain, etc. were not addressed in detail. Furthermore, we did not prepare a review on synthetic thyroid hormone analogues, which mainly act via nuclear T3 receptor (beta), nor on TH metabolites acting via the putative plasma membrane integrin receptor anb3. Thus, the reviewer’s suggestion to address interactions between muscle mass and “thyroid activity” may be beyond the scope of the 3,5-T2 focus of our review.

With respect to the in vitro data discussed only those reports using 3,5-T2 concentrations below 1-10 µM concentrations were considered, but not those studying high pharmacological 3,5-T2 concentrations, considering that endogenous 3,5-T2 concentrations probably are in the low picomolar range.
